# COVID-19 Vaccines for Optimizing Immunity in the Upper Respiratory Tract

**DOI:** 10.3390/v15112203

**Published:** 2023-10-31

**Authors:** Ranjan Ramasamy

**Affiliations:** ID-FISH Technology Inc., 556 Gibraltar Drive, Milpitas, CA 95035, USA; rramasamy@idfishtechnology.com

**Keywords:** adaptive immunity to COVID-19, clinical vaccine trials, COVID-19, COVID-19 vaccines, innate immunity to COVID-19, mucosal vaccines, nasal vaccines, SARS-CoV-2, upper respiratory tract immunity, vaccine safety

## Abstract

Rapid development and deployment of vaccines greatly reduced mortality and morbidity during the COVID-19 pandemic. The most widely used COVID-19 vaccines approved by national regulatory authorities require intramuscular administration. SARS-CoV-2 initially infects the upper respiratory tract, where the infection can be eliminated with little or no symptoms by an effective immune response. Failure to eliminate SARS-CoV-2 in the upper respiratory tract results in lower respiratory tract infections that can lead to severe disease and death. Presently used intramuscularly administered COVID-19 vaccines are effective in reducing severe disease and mortality, but are not entirely able to prevent asymptomatic and mild infections as well as person-to-person transmission of the virus. Individual and population differences also influence susceptibility to infection and the propensity to develop severe disease. This article provides a perspective on the nature and the mode of delivery of COVID-19 vaccines that can optimize protective immunity in the upper respiratory tract to reduce infections and virus transmission as well as severe disease.

## 1. Introduction

Coronavirus disease 2019 (COVID-19) was first identified in December 2019 in Wuhan, China. COVID-19 is caused by infection with the severe acute respiratory syndrome coronavirus 2 (SARS-CoV-2). The World Health Organization (WHO) recorded 771 million confirmed cases of COVID-19 and 7 million resulting deaths worldwide on 4 October 2023 [1]. Vaccines against COVID-19 were rapidly developed and then deployed from December 2020 onward. The WHO reported that 13.5 billion doses of a COVID-19 vaccine had been administered globally by 4 October 2023 [1]. Vaccination on such a large scale enhanced population immunity and minimized severe disease and mortality. During the first year of COVID-19 vaccination from December 2020 to December 2021, 55.9% of the global population were estimated to have received one dose of a vaccine, 45.5% two doses, and 4.3% a third booster immunization [2]. Based on officially reported COVID-19 deaths, 14.4 million (95% confidence interval 13.7–15.9 million), and alternatively, on the more reliable excess mortality data, 19.8 million (95% interval 19.1–20.4 million) deaths were estimated to have been averted by vaccination in a total of 185 countries over this first one-year period [2]. Vaccination also reduced SARS-CoV-2 infections and hospitalizations. In the USA, for example, one estimate suggested 8 million fewer infections, 0.7 million fewer hospitalizations, and 0.12 million less deaths in the first six months of vaccine deployment from 14 December 2020 to 3 June 2021 [3]. Comparable reductions in infections and hospitalizations with COVID-19 vaccination have been reported in other countries, e.g., Israel [4].

Different types of vaccine platforms have been used to deliver SARS-CoV-2 immunogens, and these include messenger RNA, DNA, protein subunits, viral vectors, inactivated whole virus, and virus-like particles [5,6]. The whole inactivated virus, mRNA, and non-replicating adenoviral vectors were the main types of vaccine platforms widely used with approval from pertinent regulatory authorities in most countries [5,6]. However, several other investigational COVID-19 vaccines are now in preclinical or clinical development [6,7]. The most widely used COVID-19 vaccines at the present time, and which have served to significantly reduce severe COVID-19 and mortality, are administered by intramuscular injection [5,6].

Accumulating evidence suggests that air-borne SARS-CoV-2 virions are mainly responsible for transmitting COVID-19, and that inhaled virions initially infect the nasal epithelium, followed by the nasopharynx in the upper respiratory tract (URT) [8,9,10,11]. When persons who had not previously been infected with SARS-CoV-2 or vaccinated against COVID-19 become infected, they manifest a wide range of clinical symptoms, with an estimated 20–40% being asymptomatic, while others have mild symptoms (defined here as not requiring hospitalization), and a proportion (up to an estimated 20%) develop severe disease, including pneumonia and severe acute respiratory distress syndrome [12,13]. Early and effective immune responses in the URT, which typically occur in healthy young adults and children, can limit replication of the virus and eliminate it in the URT with mild or no symptoms, thereby preventing the virus from spreading to the lower respiratory tract (LRT) to cause severe disease [8,9,10,14]. The principal effector mechanisms underlying innate and adaptive immunity to SARS-CoV-2 in the URT have been previously summarized [10].

The pathology of LRT infection by SARS-CoV-2, and the consequent severe disease that requires hospital treatment, is complex and involves a central role for dysfunctional immune responses [12,13]. SARS-CoV-2 infection of the lungs can lead to uncontrolled coagulation and inflammation in the lungs, increased permeability of vascular endothelium [15], diffuse alveolar damage where the lungs are unable to maintain adequate oxygen supply, and systemic pathology [12,13]. The primary goal of COVID-19 vaccination to date has been to prevent or minimize the development of severe disease and death, which is closely dependent on averting or rapidly eliminating SARS-CoV-2 infections in the URT. Immune responses needed in the URT for achieving this are also pertinent to two other key goals of vaccination, which are to reduce or eliminate transmission of the virus to uninfected persons [7] and maintain adequate immunity against newly evolving and rapidly spreading SARS-CoV-2 variants [7,16]. This article provides a current perspective on the types of vaccines and vaccination procedures that may have the potential to generate the desired immunity to SARS-CoV-2 in the URT.

## 2. Correlates of Protective Immunity in the URT

The factors governing protection from URT infection and severe COVID-19 involving the LRT, although closely related, are not identical [10]. Several studies have established that levels of serum antibodies that bind to the S1 domain of the SARS-CoV-2 spike protein (S), which contains the angiotensin-converting enzyme 2 (ACE2) receptor binding domain (RBD), and the in vitro virus neutralizing ability of serum antibodies are correlates of protection against the development of severe COVID-19 following infection [5,17,18]. They are also a measure of the efficacy of common intramuscularly administered COVID-19 vaccines in preventing severe disease and death [5,17,18]. Data also suggest that levels of SARS-CoV-2-specific CD4+ and CD8+ T cells in the peripheral blood circulation also correlate with similar protection [5,18]. The immune correlates of protection against SARS-CoV-2 infections occurring in the URT, on the other hand, are not well established because of (i) the greater difficulty of studying immune responses in the URT compared with blood, and (ii) infections that are eliminated in the URT with mild or no overt symptoms. However, some effector immune mechanisms functioning in the URT against SARS-CoV-2 infections have been identified [10], including evidence for antibodies against RBD and S in nasal mucosal fluid that inhibit binding to ACE2 and reduce viral loads in the URT [19,20].

### 2.1. Innate Immunity in the URT

Higher expression levels of pattern recognition receptors (PRRs) for viral RNA, types 1 and 3 interferon-stimulated genes (ISGs), inflammatory cytokines and chemokines in the URT of children, as well as early induction of ISGs in the URT of adults, correlate with milder COVID-19 infections [9,10]. Conversely, genetic defects in these and other innate immunity components in individuals increase the likelihood of more severe COVID-19 [9,10]. Genetic differences between populations also influence URT innate immune responses against SARS-CoV-2 [8,9,10], which may explain why people with a recent tropical ancestry are more prone to develop severe COVID-19 in temperate zone climates [8]. Because innate immune mediators have a role in initiating subsequent adaptive immune responses, robust innate immune responses also translate into more effective antibody and T cell responses in the URT. URT infections and nasal vaccinations may potentially produce epigenetic and functional changes in innate immune cells of the URT, a process termed trained immunity [21], that can enhance the innate immune response to a subsequent infection with SARS-CoV-2. Researchers conducting intramuscular immunization studies on mice with the S-based Pfizer/BioNTech BNT162b2 vaccine observed stronger systemic innate immune responses after a second vaccination that were consistent with systemic trained immunity [22]. Healthy children, who are less likely than adults to develop severe COVID-19, show heightened levels of basal innate immune activity in the nasal mucosa, which helps them respond more rapidly and effectively to a SARS-CoV-2 infection than healthy adults [10]. Trained immunity resulting from other frequent viral infections in the URT of children may contribute to this difference. Further investigations are therefore needed to better understand the possible contribution of trained immunity to protection against SARS-CoV-2 infections in the URT.

### 2.2. Adaptive Immunity in the URT

#### 2.2.1. After Infection with SARS-CoV-2 or Other Human Coronaviruses

An early study of 35 convalescent COVID-19 patients identified distinct differences in functional antibody responses to SARS-CoV-2 in serum and nasal washings [23]. IgA, IgG, and IgM antibodies to SARS-CoV-2 were detected in nasal washings and in sera of patients with symptoms, regardless of whether they required hospitalization or not. SARS-CoV-2 infections were found to elevate levels of antibodies cross-reacting with other types of endemic human coronaviruses in nasal washings and sera [23]. Importantly, virus neutralization assays showed that neutralization titers with IgA in nasal washings against SARS-CoV-2 were significantly higher in patients who did not require hospitalization compared with those with severe disease that did [23]. This finding is consistent with the protection afforded by mucosal IgA antibodies in other human and animal coronavirus infections [23]. IgG antibody-dependent phagocytosis by monocytes and virus neutralization with IgM antibodies were also observed in nasal washings from the same COVID-19 patients [23]. More recent findings among COVID-19 patients showed that levels of IgA antibodies to the S1 subunit in nasal epithelial lining fluid correlated with virus neutralization titers and reduced viral load in the URT [24].

Another study of COVID-19 patients admitted to hospital showed that (i) IgA antibodies to SARS-CoV-2 RBD were detected earlier in serum than IgG antibodies; (ii) a large proportion of plasmablasts synthesizing IgA in the blood express chemokine receptor 10 (CCR10), which is a marker for their homing to mucosal sites; (iii) neutralizing IgA anti-RBD antibodies were more abundant in saliva than serum, and were formed early during the onset of symptoms; and (iv) neutralizing IgA antibodies remained detectable in saliva for 49 to 73 days after symptom onset [25]. Nasal SARS-CoV-2 S-specific IgA antibody levels resulting from an infection were boosted by a subsequent infection and displayed significant variant-transcending neutralization capability against SARS-CoV-2 variants [26]. Furthermore, IgA antibodies in nasal epithelial fluid, produced after infection with early strains of SARS-CoV-2 in 2020, (i) inhibited RBD-ACE 2 binding, (ii) were significantly higher than after two intramuscular vaccinations with the mRNA1273 vaccine expressing S, (iii) recognized RBD from the Delta and Omicron BA.1 variants, and (iv) remained elevated for several months [19,20]. These observations also suggest that higher levels of antibodies in the URT, particularly antibodies of the secretory IgA type, are a likely correlate of protection in the URT.

CD4+ helper T cells (T_H_) and CD8+ cytotoxic lymphocytes (T_C_) have been implicated in protecting a group of healthcare workers lacking antibodies to SARS-CoV-2 against infection, despite their likely repeated exposure to the virus in hospitals [27]. The CD4+ T_H_ cells obtained from their peripheral blood in this study were shown to recognize cross-reactive epitopes present in replication–transcription complex proteins of SARS-CoV-2 and other common human-infecting endemic coronaviruses [27]. While this [27] and other indirect data [14,18] suggest that T_H_ and T_C_ cells are able to protect against SARS-CoV-2 infection, it has been technically difficult to obtain evidence for URT-located T cells in protection against infection. Sampling of nasal mucosal cells by nasal curettage after recovery from COVID-19 identified SARS-CoV-2-specific CD8+ T_C_ cells with a resident memory phenotype (CD8+ T_RM_) persisting in the nasal mucosa for several months after clearance of the virus [28]. A study of nasopharynx-associated lymphoid tissue (NALT) removed from children who underwent tonsillectomy and adenoidectomy, after a prior SARS-CoV-2 infection, provided important evidence for SARS-CoV-2-specific URT-resident B and T cells in the NALT [29]. This study found SARS-CoV-2-specific germinal center and memory B cells that had class switched to IgA and IgG, and undergone somatic hypermutation in the variable region genes of immunoglobulins, in NALT [29]. The B cell antigen receptor sequences in the cells were specific for S and matched known S-specific sequences identified in other studies. Tissue-resident CD4+ T_H_ cells with a memory phenotype (CD4+ T_RM_) and activated CD8+ T_C_ with T cell receptor sequences known to be specific for SARS-CoV-2 epitopes were also identified in the NALT. The T_H_ and T_C_ cells from the NALT synthesized appropriate cytokines and other proteins on in vitro activation [29]. This study [29] therefore provides evidence for the presence of tissue-resident memory B (B_RM_) and T_RM_ cells recognizing SARS-CoV-2 antigens in the URT NALT after a SARS-CoV-2 infection.

Another recent study found a significant association between asymptomatic infection with SARS-CoV-2 and a class 1 HLA allele *HLA-B*15:01* in unvaccinated persons, supporting a role for CD8+ T_C_ cells in early protection against infection, which is most likely to have occurred in the URT [30]. CD8+ T_C_ cells collected in the pre-pandemic era from *HLA-B*15:01* persons reacted with an immunodominant SARS-CoV-2 S-derived peptide epitope with the amino acid (aa) sequence NQKLIANQF. Homologous aa sequences NQKLIANQF and NQKLIANAF were present in the common cold-causing endemic coronaviruses OC43-CoV and HKU1-CoV, respectively. NQKLIANAF was shown to be presented by HLA-B*15:01 to CD8+ T_C_ cells and cross-react with the SARS-CoV-2 S epitope, thereby providing a likely explanation for pre-existing CD8+ T_C_ cell-mediated immunity against SARS-CoV-2 infection in the URT [30].

These recent findings suggest that memory B and T cells are found in URT lymphoid tissue after recovery from COVID-19 or infections with other human coronaviruses, which generate a sufficiently early protective adaptive immune response to eliminate a subsequent infection with SARS-CoV-2 without the development of symptoms. Figure 1 illustrates the main features relevant to SARS-CoV-2 infection and immunity in the URT.

#### 2.2.2. After Intramuscular Vaccination of Infection-Naïve Persons

The widely used, intramuscularly administered COVID-19 vaccines based on mRNA, replication-deficient adenovirus vectors, the whole inactivated virus, and the adjuvanted SARS-CoV-2 spike protein (S) have been shown in clinical trials to elicit virus-specific T cells and antibodies in blood, and reduce severe disease and mortality [5,10,14,18]. The protection from severe disease raises the question of whether intramuscularly administered vaccines also generate protective adaptive immunity in the URT, which then limits SARS-CoV-2 infections in the URT, thereby minimizing infection of the LRT and the development of symptomatic COVID-19. Consistent with this assertion, preclinical studies conducted on animal models with intramuscular COVID-19 vaccines have demonstrated SARS-CoV-2-specific URT immunity [10]. Experimental findings of potentially protective antibody responses in the URT elicited by the widely used intramuscularly delivered adenoviral vector, and mRNA COVID-19 vaccines expressing S in persons who had never been infected with SARS-CoV-2 (Table 1), are consistent with findings from preclinical studies.

Intramuscularly administered mRNA S-based vaccinations were found to elicit S-specific antibodies in bronchoalveolar fluid of infection-naïve vaccinees, but at lower concentrations than in COVID-19 convalescent persons [36]. URT responses were not investigated in this study [36]. Reports of T_H_ and T_C_ cell responses in the URT to intramuscularly administered adenovirus-vectored and mRNA S-based vaccines in infection-naïve persons have been sparse [10]. A recent analysis of cells obtained with nasopharyngeal swabs in persons vaccinated with the Pfizer/BioNTech S-based mRNA vaccine demonstrated the expansion of CD8+ T_RM_ cells as well as CD4+ T_H_ cells in persons with no known prior COVID-19 infection [37]. Another study, however, could not detect S-specific nasal CD4+ T_H_ and CD8+ T_C_ cells in persons receiving S-based mRNA intramuscular vaccines unless the vaccinees subsequently became infected with SARS-CoV-2 [38]. Reconciling these divergent findings is important because tissue-resident T_RM_ and B_RM_ potentially induced in the URT by intramuscular vaccination can be expected to generate the rapid anamnestic immune response required to limit SARS-CoV-2 infections to the URT, a process that commonly occurs in persons who have recovered from COVID-19 (discussed in Section 2.2.1). Studies on donated organs show that B_RM_ and T_RM_ elicited by SARS-CoV-2 infection are found in the lungs, bone marrow, multiple lymph nodes, and spleen six months after infection [39], raising the importance of investigating their presence also in the NALT and elsewhere in the URT after different types of COVID-19 vaccination, including mucosal vaccination discussed in Section 3 below. B_RM_ and T_RM_ help generate robust anamnestic immune responses in many tissues [40,41,42], and therefore, their potential presence in the URT after COVID-19 vaccination may help limit SARS-CoV-2 infections in the URT.

The kinetics of antibody and immune cell responses in the blood of persons who had been vaccinated three times with S-based mRNA vaccines, and then become infected with the Omicron variant, are also pertinent in this context [43]. Activation of S-specific CD4+ and CD8+ cells were detected during the first week of infection, together with an expansion of S-specific plasmablasts. However, a substantial increase in S-specific neutralizing antibodies occurred mainly in the second week. Antibodies to S generated after Omicron infection were predominantly directed towards epitopes shared with the parent SARS-CoV-2 strain used in the vaccines, suggestive of antigen imprinting [43]. This study [43] did not investigate URT responses, but the findings suggest that eliminating the virus early in the URT will require similar recall adaptive immune responses and adequate basal concentrations of antibodies to SARS-CoV-2, as discussed in Section 2.2.1.

Many vaccinated persons have experienced breakthrough infections from SARS-CoV-2 variants carrying multiple mutations in S that have been selected to evade neutralization with vaccine-elicited antibodies [16,18]. However, S-based booster mRNA vaccines incorporating S from the recently widespread SARS-CoV-2 variants have yielded better protection in Nordic countries [44], suggesting that this approach is also relevant for inducing URT immunity.

#### 2.2.3. After Intranasal Vaccination of Infection-Naïve Persons

Protective immunity at the site of initial infection against infecting forms of human pathogens is important for minimizing or preventing the development of disease [45,46]. Locally produced COVID-19 vaccines administered by inhalation or nasally have recently been approved for use in China, India, Iran, and Russia [7,47]. Detailed data on the immune responses elicited by these vaccines in the URT or systemically are not yet available, and their interpretation may be complicated by likely prior COVID-19 infection in many of the vaccinees. The BBV154 adenovirus-vectored nasally administered vaccine expressing S was, however, reported to induce higher salivary IgA antibody levels to S in seronegative vaccinees than an intramuscularly administered whole inactivated virus vaccine (Covaxin) in a phase 3 trial in India [48]. This finding is consistent with data showing that non-replicating adenovirus-vectored S-based vaccines delivered intranasally efficiently protect against SARS-CoV-2 infection of the URT in animal models [10,49,50].

## 3. Mucosal (Nasal and Oral) Vaccines for COVID-19

Intramuscularly administered vaccines have played a crucial role in reducing mortality and morbidity during the COVID-19 pandemic [1,2,3,4]. However, the importance of URT immune responses in limiting infections (Section 2), the need to minimize infections and transmission, the ease of administration, and the successful use of nasally administered influenza vaccines, have led to the consideration of developing mucosally administrable vaccines for COVID-19. Prospects for developing mucosally delivered vaccines, specifically through the intranasal or oral routes, for COVID-19 were discussed at a virtual workshop on 7–8 November 2022 on “Mucosal Vaccines for SARS-CoV-2: Scientific Gaps and Opportunities’’ convened by the US National Institute of Allergy and Infectious Diseases of the US National Institutes of Health. A list of some of the candidate mucosal vaccines for COVID-19 under development at the time was provided in the workshop report [7].

Oral and nasal vaccines have the advantage of developing good systemic immune responses in addition to mucosal responses. For example, antigens expressed on the cell walls of live, recombinant, food-grade lactic acid bacteria, or antigens bound to their isolated bacterial cell walls, elicit mucosal and systemic immune responses in mice and rabbits after nasal and oral immunization [51,52]. The immune response to antigens expressed by lactic acid bacteria after combined oral and nasal immunization was, however, significantly weaker in neonatal and old mice compared with young adult mice, and therefore, applying this vaccine platform to human populations requires further investigation [53]. The finding of IgA antibodies to RBD in the nasal cavity, bronchi, and gastrointestinal tract of mice after intranasal immunization with RBD displayed on the surface of live *Lactobacillus plantarum* illustrates the potential of this platform for developing a nasal COVID-19 vaccine [54]. A related approach used the edible alga *Arthrospira platensis*, expressing a protective protein antigen from a malaria parasite, administered intranasally and orally, to induce systemic antibodies that protected against a challenge malaria infection in mice [55]. Although mucosal antibodies were not measured in this study, they are likely to have been produced by the oral immunization protocol [55]. Immunization of hamsters by oral gavage with adenovirus-5 expressing S in appropriate enteric-coated pills generated S-specific IgG antibodies in blood as well S-specific IgA antibodies in the nose and oropharynx, demonstrating the potential for oral vaccination against COVID-19 [56].

### 3.1. Lessons from Influenza Vaccines for Developing Nasal COVID-19 Vaccines

While there are approved live attenuated or inactivated orally delivered vaccines for *Vibrio cholerae*, *Salmonella typhimurium*, rotavirus, and poliovirus infections in use, the only approved intranasally delivered vaccine for human use in the USA is a quadrivalent live attenuated influenza A and B vaccine [57]. The influenza A virus (IFAV) has a negative-strand RNA genome, and causes relatively mild disease when confined to the URT and more severe disease and mortality upon infection of the LRT [58,59,60]. IFAV infection is therefore a useful paradigm for SARS-CoV-2 infection [10]. Protective immune responses against symptomatic IFAV infections that are likely to eliminate infection in the URT have innate (e.g., IFNs) and adaptive (e.g., IgA and IgG antibodies, tissue-resident memory CD8+ T_C_ and CD4+ T_H_ cells) immunity components [58,59,60,61,62,63,64,65]. Nasally administered, live attenuated vaccines as well as intramuscularly administered, chemically inactivated vaccines have been used widely for many decades against influenza [58,59,65]. The composition of influenza vaccines needs to be regularly changed to accommodate mutations in the hemagglutinin and neuraminidase molecules located on the virion membrane [58,59,65]. Protection against symptomatic influenza in intramuscularly vaccinated and unvaccinated adults correlated with the levels in the peripheral circulation of (i) polyfunctional CD4+ and CD8+ T cells, including follicular helper T cells and T_H_17 cells, (ii) antibodies to hemagglutinin and neuraminidase, and (iii) myeloid dendritic and CD16+ natural killer (NK) cells [64]. The bloodstream and URT correlates of protection against the development of symptomatic influenza shared several features [58,59,60,61,62,63,64,65].

### 3.2. General Considerations for Developing Nasally Administered COVID-19 Vaccines

The desirable properties of potential nasally and orally administrable COVID-19 vaccines and the relevant immune responses discussed in Section 2 are summarized in Table 2.

A large proportion of the global population have experienced COVID-19 and/or been immunized with a COVID-19 vaccine at the present time [1]. Only very young children, who have not already been vaccinated against COVID-19, will progressively become eligible for a primary vaccination. The development of new COVID-19 vaccines and their use in vaccination programs has to take this into consideration.

Vaccines administered nasally that elicit URT, LRT, and systemic immunity constitute the simplest approach for developing more desirable COVID-19 vaccines, although several orally administered vaccines being developed may also achieve this objective [7]. The intranasal vaccines that have recently been approved for use by regulatory authorities in the respective vaccine-developing countries for this purpose are (i) an adenovirus-vectored vaccine Convidecia expressing S administered nasally by inhalation and produced by CanSinoBio in China, (ii) an adenovirus-vectored vaccine BBV154 expressing S administered nasally with a dropper and produced by Bharat Biotech in India, (iii) a combined virus-vectored vaccine GamKOVID-VacM administered nasally with a sprayer/inhaler and produced by Gamaleya in Russia, (iv) an S-based protein vaccine administered nasally by a sprayer/inhaler and produced by the Razi Institute in Iran, and (v) live attenuated SARS-CoV-2 administered nasally by a sprayer/inhaler and produced by Beijing Wantai in China [7,47].

Published data report that the BBV154 adenovirus-vectored nasally administered vaccine expressing S generated salivary IgA antibodies to S in seronegative persons, and also boosted levels of serum neutralizing antibodies to S [48]. BBV154 expressed a stabilized, prefusion form of S from the ancestral Wuhan strain of SARS-CoV-2, but intranasal vaccination elicited neutralizing antibodies against the Omicron BA.5 variant as well as the ancestral strain [48].

Despite many advantages, some potential limitations that may apply to intranasal immunizations have to be taken into consideration in developing intranasal COVID-19 vaccines. These are listed in Table 3.

### 3.3. Studies of Intranasal COVID-19 Vaccines in Animal Models

Preclinical studies have used animal models to investigate the effects intranasal vaccination after priming with intramuscular vaccines. In one study conducted on mice, priming with the mRNA BNT162b vaccine was followed by an intranasal boost with a replication-deficient adenovirus 5 vector expressing the same ancestral S. Although URT immune responses were not investigated, the intranasal boost was responsible for generating (i) systemic neutralizing antibodies, (ii) T cell immunity in the lungs, including in CD4+ cells, (iii) virus neutralizing antibodies in bronchiolar lavage fluid, and (iv) protection from severe disease and death in an intranasal SARS-CoV-2 challenge [70].

Another COVID-19 intranasal vaccine platform recently shown to generate a promising mucosal antibody response was an inactivated whole virus without exogenous adjuvant that elicited nasal wash IgA and serum IgG in mice [71]. The absence of an adjuvant reduces the risk of possible adverse events listed in Table 3. Intranasal vaccination was more protective in the URT than subcutaneous administration of inactivated virus, and importantly, the intranasal vaccination boosted anti-S IgA in the nasal mucosa and anti-S IgG in the blood following a priming intramuscular S-expressing mRNA vaccination in this study [71]. Because nasal IgA antibodies protect against URT infection, and the world’s population has largely been primed against S already, these findings are an important consideration for developing nasal COVID-19 vaccines.

Recombinant RBD, which was linked C terminally to a bacterial protein that binds to mucosal microfold cells in the URT, and adjuvanted with a strong toll-like receptor 3 (a PRR) agonist, has been shown to elicit specific IgA and IgG antibodies in the URT, LRT, and blood, as well as systemic CD4+ and CD8+ T cells [72].

The avian paramyxovirus type 3 (APMV3) virus, which is not infective to humans, was utilized to express S in a prefusion-stabilized conformation, and then used to immunize hamsters [73]. A single intranasal vaccination elicited (i) high levels of neutralizing IgG and IgA in serum, (ii) detectable variant-transcending neutralization, and (iii) undetectable or low replication of challenging SARS-CoV-2 in the URT and LRT. The findings were considered sufficiently promising to progress this vaccine formulation to clinical trials [73].

A related effort utilized a live attenuated recombinant bovine/human parainfluenza-virus-vectored vaccine candidate expressing SARS-CoV-2 prefusion-stabilized S termed B/HPIV3/S-6P [74]. B/HPIV3 had previously been tested as a pediatric vaccine for children under five years old with a good safety profile. The immunogenicity and protective efficacy of B/HPIV3/S-6P was tested in rhesus macaques. A single intranasal/intratracheal dose of B/HPIV3/S-6P induced strong S-specific IgA and IgG antibodies in the URT and LRT, as well as serum antibodies that were able to neutralize the infectivity of the ancestral SARS-CoV-2 strain from which the immunizing S was derived, as well several variants [74]. B/HPIV3/S-6P also induced systemic and pulmonary S-specific CD4+ and CD8+ T cells responses, including T_RM_ cells in the lungs. SARS-CoV-2 replication was not detectable in the URT and LRT of immunized macaques in a challenge infection. Clinical trials on the use of B/HPIV3/S-6P as a combined vaccine for parainfluenza type 3 and COVID-19 are planned in children, based on the promising findings in macaques [74].

A different approach aiming to improve upon existing intramuscular vaccines, and build on their efficacy in inducing systemic immune responses that protect against severe disease, used mRNA expressing S from an early SARS-CoV-2 strain (Pfizer BioNTech 162b2) for intramuscular priming, and unadjuvanted S administered nasally as a boosting immunization in mice [75]. Priming alone with Pfizer BioNTech 162b2 generated low levels of IgA and IgG antibodies against S in the blood. Only the combination of this priming and boosting immunizations and neither alone, led to high levels of IgA and IgG antibodies in nasal wash and bronchiolar lavage fluids [75]. Similarly, only the combination of priming and boosting immunizations led to (i) a significant increase in the levels of IgA and IgG antibodies to S in blood, (ii) S-specific B_RM_ class switched to be able to produce IgA and IgG in the lungs, (iii) S-specific CD4+ and CD8+ T_RM_ in the lungs and bronchiolar lavage fluid, (iv) decreased virus levels in the URT and LRT, and complete protection from severe disease and death, when immunized transgenic mice expressing the human ACE2 receptor were lethally challenged intranasally with SARS-CoV-2, and (v) the generation of CD4+ T_H_1 and T_H_17 T_RM_ subsets desirable for protective T cell-mediated immunity in the lungs [75].

Furthermore, this priming and boosting combination was also effective in similarly protecting hamsters from disease, reducing their shedding of virus, and also diminishing their URT infections when placed in close contact with infectious hamsters [75]. Additionally, when the intranasal boosting immunization was carried out with S protein derived from SARS-CoV-1, mucosal and systemic antibody and cellular responses were generated against S from both SARS-CoV-2 and SARS-CoV-1, suggesting that this intramuscular prime and intranasal boost approach can be successfully used to generate immunity to evolving SARS-CoV-2 variants [75]. Therefore, the intramuscular prime and intranasal boost approach seems to meet many of the desirable characteristics of protective immunity in the URT and the LRT outlined in Table 2.

## 4. Conclusions

Clinical safety and immunogenicity trials with new vaccine platforms being developed for nasal and oral immunizations need to carefully consider possible limitations that may specifically apply to them (Table 3). Vaccine trial protocols developed with a focus on safety elements are likely to be useful for this purpose [76]. The progress made in elucidating the mechanisms of immunological protection against SARS-CoV-2 infections and the rapid development of new vaccine platforms suggests that safe and even more effective COVID-19 vaccines may become available in the near future.

## Figures and Tables

**Figure 1 viruses-15-02203-f001:**
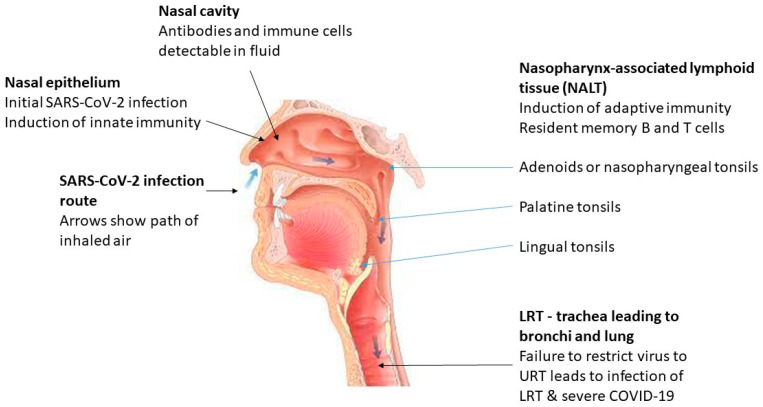
Infectivity and immunity of SARS-CoV-2 in the URT. Modified from reference [10] with permission under the creative commons attribution (CC BY) license.

**Table 1 viruses-15-02203-t001:** Antibody responses in the URT of SARS-CoV-2 infection naïve persons after intramuscular vaccination with S-based adenovirus-vectored and mRNA vaccines.

Vaccine	Immune Response	Reference
1. ChAdOx1 nCoV-19 (AZD1222)—replication-deficient simian adenovirus expressing S	Anti-S IgG antibodies in nasal fluid, probably translocated from plasma by neonatal Fc receptors, persisting for a year after boost	[31]
2. Pfizer/BioNTech BNT162b2 and Moderna mRNA1273 mRNA vaccines expressing S	Anti-S IgG and IgA antibodies in saliva and nasal fluid	[19,20,32,33,34,35]

**Table 2 viruses-15-02203-t002:** Desired properties of mucosally administered COVID-19 vaccines.

Desired Property	Associated Immune Responses
1. Prevent or minimize infection of the URT (diminished virus shedding and person-to-person transmission are dependent on this)	Elicit and maintain adequate levels of protective antibodies as well as memory B and effector memory T cells in the URT mucosa and NALT. Facilitate effective innate immunity
2. Prevent or minimize LRT and systemic infection (infections of the LRT and more systemic spread of virus are always possible even with good URT immunity)	(i) Generate and maintain adequate levels of pre-existing antibodies as well as memory B and effector memory T cells in the LRT and associated lymphoid tissue and lymph nodes. Early and effective innate immune response in the LRT (ii) Produce and maintain adequate blood levels of antibodies and effector T cells as well as memory B and T cells in lymphoid organs
3. Vaccine platform readily adaptable to protect against infection with newly emerging variants of SARS-CoV-2	Similar immune responses to the variants as described in (1) and (2) above with minimal antigen imprinting effects in B and T cells
4. Safety and efficacy over a wide age range	(i) Effective immune responses as in (1) and (2) in children and young adults as well as the elderly (ii) Absence of possible adverse effects for nasal vaccines listed in Table 3

**Table 3 viruses-15-02203-t003:** Potential general limitations with intranasal vaccines.

Possible Limitation	Reference
Type 1 hypersensitivity reaction to antigenic molecules reaching lungs	[66]
Exacerbation or development of airway diseases, e.g., asthma and rhinitis	[66]
Adverse neurological events	[67,68]
Possible disseminated infections with attenuated vectors and viruses in immunocompromised persons	[69]

## Data Availability

All data supporting the conclusions of this article are included within the article.

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
