# Peer review of "COVID-19 Vaccines for Optimizing Immunity in the Upper Respiratory Tract"

_viruses, 2023, doi:10.3390/v15112203_

Round 1

Reviewer 1 Report

Comments and Suggestions for Authors

The article is well written, reviewing the important aspects regarding vaccines to prevent Covid, only one aspect that could be improved is to add a table with the properties desired in mucosally-administered vaccines referred in line 294-295:

Line 294-295 A recently published report of this workshop summarized the properties desired in mucosally-administered vaccines for COVID-19 and provided a list of candidate mucosal vaccines under development.

Author Response

I am grateful for this helpful comment. 

A new Table 2 on desired properties of mucosal vaccines has been added, and the reference to NIH report [which did not give details of this] has been modified in limes 296-298 accordingly. The changes made in the revision have been highlighted.  

Reviewer 2 Report

Comments and Suggestions for Authors

The manuscript by Ranjan Ramasamy is a review providing a perspective on the nature and the mode of delivery COVID-19 vaccines. It is focused on the introduction of optimized formulations able to increase the upper respiratory tract protective immunity. The use of intramuscularly COVID-19 vaccine administration is effective in reducing severe disease and mortality but is limited is preventing asymptomatic and mild infections, and person to person transmission.

The review is very well presented, data reported are based on appropriate literature, and I think it can strongly contribute to the scientific progression in the field of COVID-19 vaccines.

I suggest just a point that can be better detailed in the text: the possible limitation in the generation/boosting of immune response by mucosal vaccines for old people and children compared to intramuscular vaccine administration.

Author Response

I am grateful for this helpful comment.

The relevant point is now discussed in the text in lines 304-310 and not in the relevant Table [now Table 3].

The changes have been highlighted in the revised manuscript.